# The blue carbon of southern southwest Atlantic salt marshes and their biotic and abiotic drivers

Paulina Martinetto [1] ✉, Juan Alberti[1], María Eugenia Becherucci[1], Just Cebrian [2,9], Oscar Iribarne[1], Núria Marbà [3], Diana Montemayor[1], Eric Sparks[4,5] & Raymond Ward[6,7,8]

Coastal vegetated ecosystems are acknowledged for their capacity to sequester organic carbon (OC), known as blue C. Yet, blue C global accounting is incomplete, with major gaps in southern hemisphere data. It also shows a large variability suggesting that the interaction between environmental and biological drivers is important at the local scale. In southwest Atlantic salt marshes, to account for the space occupied by crab burrows, it is key to avoid overestimates. Here we found that southern southwest Atlantic salt marshes store on average 42.43 (SE = 27.56) Mg OC·ha$^{-1}$ (40.74 (SE = 2.7) in below-ground) and bury in average 47.62 g OC·m$^{-2}$·yr$^{-1}$ (ranging from 7.38 to 204.21). Accretion rates, granulometry, plant species and burrowing crabs were identified as the main factors in determining belowground OC stocks. These data lead to an updated global estimation for stocks in salt marshes of 185.89 Mg OC·ha$^{-1}$ ($n = 743$; SE = 4.92) and a C burial rate of 199.61 g OC·m$^{-2}$·yr$^{-1}$ ($n = 193$; SE = 16.04), which are lower than previous estimates.

Over the last decade, coastal vegetated areas (e.g., salt marshes, mangroves, seagrass meadows) have been recognized for their high capacity to sequester carbon[1]. It is under the climate change context that the term 'Blue carbon' emerged to refer to the organic carbon fixed by coastal vegetation that can be stored for centuries to millennia[2]. Conservation and restoration of blue C ecosystems have been recognized as a need to mitigate and adapt to climate change[3,4]. Thus, a range of studies have documented blue C stocks around the world, in many cases as part of national C emission accounting and reduction frameworks[5–8].

At a global scale, salt marsh blue C accounting is still incomplete and is heavily skewed toward studies from the northern hemisphere.

Besides a recently growing availability of data from Australia[9], southern hemisphere data is scarce. In particular, for South America, a recent revision[10] makes evident the limited information that exists for salt marshes, with most of the estimation derived from the percentage of organic matter in surficial soils. As more data becomes available, the huge variability in C stocks and sequestration rates has become clearer, not just among different vegetation types but also within a given ecosystem. For instance, in a study of 84 Australian salt marshes, C stocks varied from 14 to 962 Mg C ha$^{-1}$ in the top 1 m[9] and a global analysis showed sequestration rates in salt marshes ranges from 18 to 1713 g C·m$^{-2}$·year$^{-1}$[11]. Findings like these suggest that the interaction among environmental variables and the ecological mechanisms that determine

[1]Laboratorio de Ecología, Instituto de Investigaciones Marinas y Costeras (IIMyC, UNMdP-CONICET), Juan B Justo 2550, Mar del Plata (7600), Argentina. [2]Northern Gulf Institute, Mississippi State University, NOAA NCEI, 1021 Balch Blvd, Stennis Space Center, MS 39529, USA. [3]Global Change Research Group, IMEDEA (CSIC-UIB), Institut Mediterrani d'Estudis Avançats, Miquel Marquès 21, 07190 Esporles, Illes Balears, Spain. [4]Coastal Research and Extension Center, Mississippi State University, 1815 Popp's Ferry Rd., Biloxi, MS 39532, USA. [5]Mississippi-Alabama Sea Grant Consortium, 703 East Beach Drive, Ocean Springs, MS 39564, USA. [6]School of Geography, Queen Mary University of London, Mile End Rd, Bethnal Green, London E1 4NS, United Kingdom. [7]Institute of Agriculture and Environmental Sciences, Estonia University of Life Sciences, Kreutzwaldi 5, EE-51014 Tartu, Estonia. [8]Colégio de Estudos Avançados, Universidade Federal do Ceará, Campus do Pici, CEP 60455-760 Fortaleza, CE, Brasil. [9]Present address: "Vesta, PBC", 584 Castro St, #2054, San Francisco, CA 94114-2512, USA. ✉e-mail: pmartin@mdp.edu.ar

blue C are important at the local scale and highlight that global estimates derived from indirect methods should be taken with caution.

Several environmental variables have been proposed to explain variability in blue C. Dominant species, local geomorphology, nutrient availability, hydroperiod, salinity and suspended sediment supply are among the most studied[9,11–14]. Previous efforts have often failed to include biological processes, even though these processes have long been recognized as important controllers of C transformation pathways. Bioturbation and herbivory, for instance, are important processes that affect C cycling with multiple antagonistic and synergistic effects[15–17]. Specifically, crab bioturbation and herbivory, through a complex network of positive, negative, direct and indirect interactions, can profoundly affect C transformation pathways in salt marshes of the southern southwest (SW) Atlantic coast[15]. Bioturbation and herbivory may be overlooked processes in studies of C stocks and sequestration rates in salt marshes around the world, and even other blue C ecosystems, given that coastal vegetated ecosystems are usually inhabited by burrowing organisms that consume plants and disturb the sediment[18–20]. Furthermore, through bioturbation, infaunal organisms can shape marsh geomorphology by, for instance, facilitating creek formation[21] or stimulating landscape-scale accretion[22].

To address the knowledge gaps, we carry out a comprehensive study of C stocks and burial rates in southern SW Atlantic salt marshes covering a latitudinal gradient from 35° to 51° S. In addition, to test the importance of selected biological and environmental variables in determining the C stocks of the region, we carried out a path analysis using SEM (structural equation modeling). Finally, we updated global estimates of C stocks and burial rates for salt marshes.

## Results and discussion
### Southern SW Atlantic salt marsh organic C stocks
Most of the salt marshes in South America are located on the Atlantic coast, with over 95% of the surface of these Atlantic marshes located in

Argentina (209,056 ha from 218,964)[15,23]. In this study, we sampled 11 salt marshes covering 197,577 ha, of which 145,309 ha are dominated by strictly salt marsh species (*Spartina densiflora*, *Spartina alterniflora* and *Salicornia* sp.) and 52,268 ha by brackish water species[15]. All the results shown here correspond to the area dominated by salt marsh species. These sites span along ~3000 km of coastline (35° to 51° S, Fig. 1).

Total organic C (OC) stocks (i.e., above and belowground pooled together) ranged from 6.47 (SE = 1.5) Mg OC·ha⁻¹ in the *Salicornia* sp. salt marsh in Caleta Los Loros to 111.92 (SE = 7.2) Mg OC·ha⁻¹ in the *Spartina alterniflora* salt marsh in Riacho San José, averaging 42.43 (SE = 27.56) Mg OC·ha⁻¹ across all marshes examined (Table 1, Fig. 1). OC stocks in salt marshes dominated by *S. densiflora* were less variable than those dominated by *S. alterniflora* or *Salicornia* sp. (Table 1, Fig. 1). In general, the greater part of the total OC stock was located belowground. The contribution of the root OC stock to the total stock could reach up to 40% in *S. alterniflora* and *Salicornia* sp. salt marshes, especially in those located in the southern sites. In contrast, the contribution of the root OC stock to the total stock was less than 10% in *S. densiflora* salt marshes, with the exception of Mar Chiquita and Río Negro where it was 11.6% for both of them (Table 1, Fig. 1).

The senescent aboveground biomass OC stock contributed less than 2% to the total stock in all *S. alterniflora* and *Salicornia* sp. salt marshes, while in *S. densiflora* salt marshes, the contribution of this stock was much higher, reaching values of 11.5, 9 and 7.7 % in Riacho San José, Río Negro and Bahía Samborombón respectively (Table 1, Fig. 1). The contribution of the green aboveground biomass OC stock was in general less than 10%, with the exceptions of the Caleta Los Loros *Salicornia* sp. salt marsh (11.1%) and *S. densiflora* salt marshes in Río Negro and Riacho San José (11.1 and 10.4% respectively; Table 1, Fig. 1).

The mean total OC stock for all the sites sampled was 42.43 (SE = 27.56) Mg OC·ha⁻¹, with a mean belowground stock of 40.74

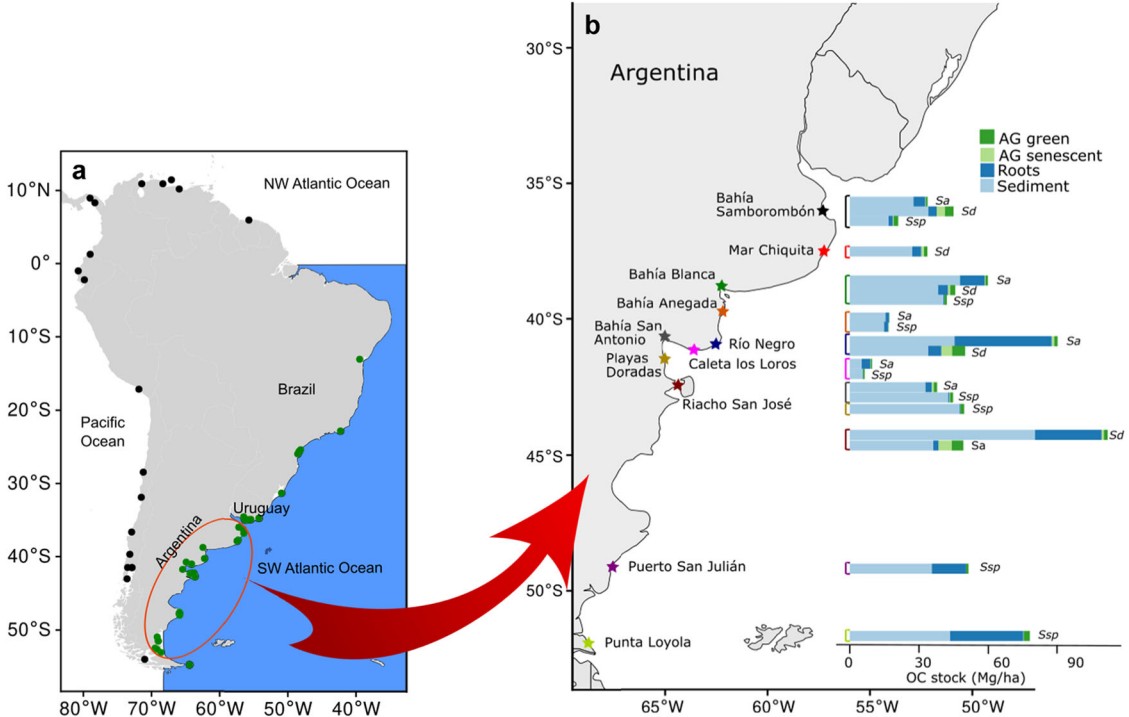

**Fig. 1 | Southern SW Atlantic salt marshes organic carbon stocks. a** South America map[60] showing sites where salt marshes occur described using black and green points. Green points indicate salt marshes in the SW Atlantic Ocean. Red circle covers the study area. **b** South American Atlantic coast with salt marsh sites included in this study indicated with colored stars. Inset: organic carbon (OC) stocks measured at each salt marsh by plant species and stock allocation. AG: aboveground, *Sa*: *Spartina alterniflora*, *Sd*: *Spartina densiflora*, *Ssp*: *Salicornia* sp. (Source data are provided as a Source Data file).

**Table 1 | Crab burrow abundances and entrance diameter, salt marsh plant biomasses, sediment sample depths and organic carbon (OC) stocks in plant biomass and in sediment for each site and dominant plant species**

| Site | Plant species | Core depth | Crab burrows | | Plant biomass | | | OC stock | | | | |
|---|---|---|---|---|---|---|---|---|---|---|---|---|
| | | | Burrow abundance | Burrow diameter | Roots | AG green tissue | AG senescent tissue | Roots | AG green tissue | AG senescent tissue | Belowground | Total |
| Bahía Samborombón | Sa | 74.4 (8.7) | 4 (5.9) | 1.5 (0.9) | 2099.4 (981.3) | 207.3 (56.5) | 72.4 (42.5) | 5.18 (2.3) | 0.81 (0.2) | 0.20 (0.1) | 32.81 (6.8) | 33.82 (6.9) |
| | Sd | 49.4 (0.7) | 4.4 (0.4) | 1.9 (0.1) | 1215.5 (305.1) | 913.5 (198.1) | 848.6 (295.2) | 3.64 (1.1) | 3.85 (0.7) | 3.45 (1.0) | 37.79 (8.0) | 45.10 (8.3) |
| | Ssp | 56.8 (3.2) | 6.8 (0.6) | 1.7 (0.2) | 729.3 (223.1) | 586 (83.6) | 59.3 (31.2) | 1.84 (0.6) | 2.08 (0.2) | 0.23 (0.1) | 18.71 (1.9) | 21.02 (2.0) |
| Mar Chiquita | Sd | 79.4 (4.3) | 6.6 (1.2) | 2.1 (0.1) | 1107 (585.2) | 363.3 (87.4) | 292.2 (78.5) | 3.90 (1.9) | 1.55 (0.4) | 1.14 (0.3) | 31.05 (3.4) | 33.74 (3.2) |
| Bahía Blanca | Sa | 100 (0) | 5 (2) | 20.4 (0.1) | 3931.5 (659.5) | 317.3 (56.6) | 66 (18.6) | 10.81(1.8) | 1.18 (0.2) | 0.22 (0.1) | 58.62 (5.1) | 60.03 (5.0) |
| | Sd | 100 (0) | 13.8 (1.3) | 2.1 (0.1) | 1505 (390.8) | 542.9 (110.4) | 253.5 (120.5) | 4.38 (1.2) | 2.19 (0.5) | 0.99 (1.4) | 42.67 (1.9) | 45.85 (1.7) |
| | Ssp | 100 (0) | 1.6 (0.9) | 1.8 (0.3) | 126.5 (50.4) | 324.2 (47.9) | 3.5 (2.6) | 0.38 (0.1) | 1.09 (0.2) | 0.01 (0.0) | 41.10 (4.9) | 42.20 (5.0) |
| Bahía Anegada | Sa | 50 (0) | 0.2 (0.2) | 4.7* | 461.1 (9184.9) | 45.5 (13.2) | 9.3 (3.1) | 1.56 (0.7) | 0.16 (0.4) | 0.02 (0.0) | 17.08 (1.5) | 17.26 (1.6) |
| | Ssp | 50 (0) | 0 | 0 | 245.7 (167.4) | 71.7 (29.1) | 11.7 (16.5) | 0.83 (0.4) | 0.14 (0.8) | 0.04 (0.03) | 15.21 (8.9) | 15.38 (1.6) |
| Bahía San Antonio | Sa | 50 (0) | 7 (1.4) | 1.9 (0.3) | 780.1 (299.5) | 436.6 (75.7) | 236.2 (18.7) | 2.29 (0.7) | 1.54 (0.4) | 0.80 (0.2) | 36.64 (2.2) | 38.98 (2.3) |
| | Ssp | 36 (0.3) | 0.2 (0.2) | 2* | 170.9 (62.2) | 444 (100.3) | 43.3 (25.3) | 0.62 (0.2) | 1.35 (0.2) | 0.16 (0.01) | 43.31 (5.4) | 44.82 (5.6) |
| Río Negro | Sa | 100 (0) | 12.7 (1.5) | 1.6 (0.1) | 10,276.2 (3415.6) | 470.7 (19.3) | 294.1 (69.3) | 33.88 (9.1) | 1.54 (0.2) | 1.04 (0.2) | 82.91 (16.8) | 85.5 (13.9) |
| | Sd | 39.3 (2.9) | 1.7 (1.1) | 1.4 (0.2) | 1620.6 (180.4) | 13,65.9 (339.7) | 1187 (273.9) | 0.66 (0.5) | 5.71 (1.2) | 4.64 (0.6) | 39.06 (3.9) | 49.41 (4.3) |
| Caleta Los Loros | Sa | 50 (0) | 3.6 (1.3) | 2.6 (0.2) | 1458.6 (208.2) | 195.5 (29.3) | 65.8 (15.7) | 33.79 (0.7) | 0.61 (0.1) | 0.14 (0.0) | 8.99 (1.1) | 9.73 (1.17) |
| | Ssp | 50 (0) | 0 | 0 | 61.7 (12.9) | 262.2 (71.3) | 3 (3.6) | 0.19 (0.05) | 0.71 (0.2) | 0.01 (0.0) | 5.74 (1.4) | 6.47 (1.5) |
| Playas Doradas | Ssp | 64.6 (6.9) | 10 (2.3) | 1.9 (0.1) | 117.7 (37.2) | 542.2 (144.3) | 0 | 0.36 (0.1) | 1.65 (0.5) | 0 | 48.09 (8.7) | 49.74 (8.5) |
| Riacho San José | Sa | 100 (0) | 13.6 (2.8) | 2.3 (0.2) | 9770.2 (2070.3) | 406.2 (54.9) | 270.6 (68.9) | 29.04 (0.6) | 1.73 (0.2) | 0.92 (0.2) | 108.81 (7.0) | 111.92 (7.1) |
| | Sd | 55 (6.1) | 0 | 0 | 661.2 (314.4) | 1279 (385.2) | 1526.7 (226) | 2.37 (0.9) | 5.15 (1.1) | 5.66 (0.7) | 38.57 (1.2) | 49.38 (7.2) |
| Puerto San Julián | Ssp | 72 (10.8) | 0 | 0 | 4530.4 (1540.5) | 378.9 (98) | 0.44 (0.5) | 14.9 (5.1) | 1.04 (0.3) | 0.02 (0.0) | 50.55 (1.1) | 51.59 (11.6) |
| Punta Loyola | Ssp | 67.6 (9.8) | 0 | 0 | 11,809.9 (1764.4) | 990.7 (177.1) | 18.44 (20.62) | 31.88 (4.6) | 2.74 (0.8) | 0.03 (0.0) | 75.38 (9.7) | 78.14 (9.7) |

Values are mean (SE). Burrow diameter (cm) and abundance (in 25 × 25 cm quadrats) and core depths are in cm; plant biomasses are in g·m²; OC stocks are in MgOC·ha⁻¹. Belowground stocks include all the OC found in soils (sediment and roots). Total OC stocks include the stocks in green and senescent tissues and belowground stocks.

AG aboveground, Sa *Spartina alterniflora*, Sd *Spartina densiflora*, Ssp *Salicornia* sp.

*Only one burrow found and measured (Source data are provided as a Source Data file).

(SE = 2.70) Mg OC·ha$^{-1}$. Altogether, the 11 salt marsh sites in this study contain 4,136,190 tons of OC, of which 3,928,493 tons were found belowground (95%, Table 2, Fig. 1).

### Southern SW Atlantic salt marsh accretion and organic C burial rates

Sediment accretion rates ranged from 0.53 mm·yr$^{-1}$ in the lower salt marsh of Riacho San José to 2.86 mm·yr$^{-1}$ in the upper salt marsh of Bahía San Antonio. C burial rates ranged from 7.38 g OC·m$^{-2}$·yr$^{-1}$ in the lower part of Playas Doradas to 204.21 g OC·m$^{-2}$·yr$^{-1}$ in the lower part of Río Negro (Table 2). Altogether, the sites examined in this study have the potential to sequester 55,100.1 tons of OC per year.

### Autochthonous and allochthonous contribution to sediment organic C stocks

C and N isotopic signatures differed among plant species, plant parts, and sites. In general, *Salicornia* sp. was C-depleted compared with *Spartina* spp. and litter was N-depleted compared with live vegetation (green leaf and roots) (Supplementary Table 1). Particulate organic matter (POM) was the main source of OC to the sediment of Bahía Samborombón, Bahía Blanca and Bahía Anegada in areas dominated by *S. alterniflora*, contributing 81, 77 and 93% of the OC, respectively. Similarly, POM was the main source of OC in Mar Chiquita, where *S. densiflora* dominated the salt marsh, contributing 80% of the OC, as well as in Bahía Blanca and Bahía Anegada in areas dominated by *Salicornia* sp., with a contribution of 89 and 90% of the C, respectively (Supplementary Table 2). All other sites had sediments with a mixture of OC derived from POM and plants, as well as macroalgae in the cases of Bahía San Antonio and Puerto San Julián.

According to these results, allochthonous POM is a major source of OC in southern SW Atlantic salt marshes. This finding suggests these ecosystems act as traps for allochthonous OM. Several studies have shown that the main sources for OC stocks in blue C ecosystems are allochthonous[24,25], suggesting that the vegetation structure in such ecosystems can play a significant role in retaining exogenous OM. Crab burrows can act as passive traps capturing organic matter and sediments imported from adjacent systems through tides and winds[26–28], given that we have identified crab burrows in the path analysis as a driver of belowground OC stocks, it could partly explain the mechanism that influences the principal origin of the stored OC.

### Biotic and abiotic drivers of belowground organic C stocks

Path analysis of the environmental (i.e., abiotic and biotic factors) influence on belowground OC stocks showed a good fit (Fisher's C = 125.794, P = 0.438, df = 124), and revealed that a combination of abiotic and biotic factors explained a considerable amount of the variability observed in belowground OC stocks (marginal R-squared = 0.47, conditional -i.e., including idiosyncratic differences between sites- R-squared = 0.84). The main direct drivers of belowground OC stocks were sediment accretion rates, grain size, crab burrows and dominant plant species (Fig. 2). Aboveground biomass and mean annual temperature were also retained since, although not significant (P = 0.103 and P = 0.116 respectively), their exclusion would have made the marginal R-square value fall to 0.25. Surprisingly, most significant effects were direct, with indirect effects being nonsignificant or small and often from variables that also affect belowground OC stocks directly (i.e., mean annual temperature and crab burrows through aboveground biomass). While different combinations of environmental variables affected the biotic components of the metamodel, these biotic components had a minor role in OC stocks. Only crab burrows and the dominant plant species (that also harbors differences in unmeasured abiotic variables) affected OC stocks, with aboveground biomass being marginally nonsignificant.

Sediment accretion rates (SAR) and grain size have been previously identified as drivers of coastal OC stocks with the general trend of larger stocks in finer sediments[13,29]. SAR may have counter effects on OC stock; while it has been hypothesized that higher accretion rates could help bury organic C and therefore increase the stocks, it also could have a dilution effect if the ratio between organic matter and sediments is too low[26,30]. Our path analysis shows that both grain size and SAR are important drivers of OC belowground stocks in southern SW Atlantic salt marshes, with a general positive effect of SAR indicating that higher rates contribute to increased stocks.

Biotic factors have received much less attention. The ecological functioning of SW Atlantic salt marshes has been well studied (e.g., ref. 31), and it has been previously suggested that burrowing crabs could have an important role in influencing OC stocks[15]. Several mechanisms driven by burrowing crabs affect the OC cycle in SW Atlantic salt marshes, either through herbivory or bioturbation. These mechanisms can have a positive or negative impact on the stocks, but overall, our path analysis revealed a general positive effect. Such positive effects may stem from detritus trapping in crab burrows[27,29], facilitation of arbuscular mycorrhizal fungi associations, which increase plant productivity[32,33], and burial of dead plant biomass and imported organic matter through sediment reworking for burrow maintenance[34]. However, many studies suggest a negative impact of crabs on belowground OC stocks in salt marshes. For instance, $CO_2$ and $CH_4$ effluxes are enhanced by the presence of crab burrows[17]. Burrows also promote carbon-rich fluid exchange between marsh and tidal creeks and increase oxygenation deep in the sediment, reducing the capacity for C sequestration[35]. In SW Atlantic salt marshes, plant consumption by crabs[36,37] may further damage the leaves through increased fungal infection and lower plant productivity[38,39]. This may also result in large inputs of senescent biomass, which may increase aerobic decomposition in the sediment[40]. Furthermore, by disturbing sediment during burrow maintenance, crabs enhance erosion at the marsh edge[27] and facilitate the creation of new tidal creeks[21].

According to our path analysis, dominant plant species was also a driver of belowground OC stocks. This analysis provides an integrative picture highlighting the main factors that determine belowground OC stocks in southern SW Atlantic marshes. To fully understand how plant species drive OC stocks, it is necessary to consider a complex set of environmental conditions behind the dominant species identity. In particular, the plant species that dominate the SW Atlantic salt marsh present two different growth forms: *Spartina* species are grasses, and *Salicornia* sp. is a succulent. Given that the establishment of a dominant species is the result of biological interactions under specific abiotic conditions, it is important to recognize that the identity of a dominant species implies much more than a taxonomic description for explaining belowground OC stocks and probably includes an entire set of environmental conditions not fully measured here.

### Updating global salt marsh blue C estimates

To compare OC stocks from this study to those reported elsewhere, we focus on belowground (roots and sediments) stocks, given that very few studies have reported aboveground stocks. OC stocks in soils have been estimated to hold an average of 99.2% of the total stock[41], although the data in this study show a lower percentage (84.9%). We recalculate the global belowground C stock average pooling our results with the data compiled by Alongi[42] and from recent publications (Source data are provided as a Source Data file). The updated global average C stock for salt marshes was 185.88 (SE = 4.92) Mg OC ha$^{-1}$. The same procedure was used to recalculate global C accumulation rates. When we recalculate the global average accumulation rate, adding our results with the data from the publications mentioned above, we find a value of 199.61 (SE = 16.04) gOC m$^{-2}$ year$^{-1}$ (Source data are provided as a Source Data file).

## The blue C of the Southern SW Atlantic salt marshes in the global context

Global estimations for salt marshes have varied as more data have become available, with a current average range of 162 Mg OC ha$^{-1}$[1] to 317.2 Mg OC ha$^{-1}$[42]. However, in general, recent publications report average OC stocks below the Alongi[25] estimate (e.g., 165, 62.5, 82, 38, 65, 224 and 207 Mg OC ha$^{-1}$ from Australia, Florida [USA], the Arabian Gulf, China, Canada and South Africa, respectively[9,43–47]).

**Table 2 | Total organic carbon (OC) stocks, accretion rates and carbon burial rates for the southern SW Atlantic coastal sites with salt marshes included in this study**

| Site | Latitude | Area (ha) | Plant species | Total OC stocks (tons) | Total OC stocks belowground (tons) | Accretion rates (mm·yr$^{-1}$) | OC burial rates (gC·m$^{-2}$·yr$^{-1}$) | Total OC burial rates (tC·yr$^{-1}$) | Main drivers of salt marsh loss |
|---|---|---|---|---|---|---|---|---|---|
| Bahía Samborombón | 35° 13'–36° 18' | 39,710 | Sa, Sd, Ssp | 1,100,351 | 973,364 | 0.96 LS 1.16 US | 37.84 LS 29.90 US | 12,274.54 | Fire and cattle, invasive species, SLR and storm surges |
| Mar Chiquita | 37° 29'–37° 46' | 3882 | Sd | 130,974 | 120,529 | NA | NA | NA | Fire and cattle, invasive species, SLR and storm surges |
| Bahía Blanca | 38° 41'–39° 30' | 29,634 | Sa, Sd, Ssp | 1,414,699 | 1,379,077 | 1.03 LS 0.65 US | 28.60 LS 18.23 US | 8467.23 | SLR, landfill, eutrophication |
| Bahía Anegada | 39° 48'–40° 42' | 62,563 | Sa, Ssp | 1,000,852 | 989,922 | 0.66 LS 1.24 US | 8.45 LS 56.21 US | 27,007.98 | Invasive species |
| Río Negro | 41° 00' | 703 | Sa, Sd | 36,434 | 29,523 | 3.44 LS 1.51 US | 204.21 LS 65.90 US | 528.28 | SLR |
| Caleta de los Loros | 41° 01' | 470 | Sa, Ssp | 4477 | 4126 | 0.67 LS 0.62 US | 7.38 LS 10.30 US | 35.55 | SLR |
| Bahía San Antonio | 40° 42'–40° 50' | 4192 | Sa, Ssp | 175,814 | 167,761 | 0.63 LS 2.86 US | 15.32 LS 58.36 US | 1556.31 | Landfill, SLR |
| Playas Doradas | 41° 36' | 30 | Ssp | 1481 | 1432 | 0.80 LS 0.58 US | 31.53 LS 13.14 US | 9.39 | SLR |
| Riacho San José | 42° 24' | 356 | Sa, Sd | 13,224 | 12,638 | 0.53 LS 0.95 US | 22.91 LS 11.02 US | 52.08 | SLR |
| Puerto San Julián | 49° 16' | 1369 | Ssp | 70,630 | 69,209 | 1.48 LS 0.86 US | 76.59 LS 50.25 US | 1048.54 | SLR |
| Punta Loyola | 51° 37' | 2400 | Ssp | 187,557 | 180,912 | 0.84 LS 1.63 US | 34.56 LS 171.67 US | 4120.20 | SLR |

Total C stocks include above and belowground stocks.

Area includes only the area occupied by the three dominant species (Source data are provided as a Source Data file).

*NA* data not available, *SLR* sea level rise, *LS* lower salt marsh, *US* upper salt marsh. Plant species are: *Sa Spartina alterniflora, Sd Spartina densiflora, Ssp Salicornia* sp.

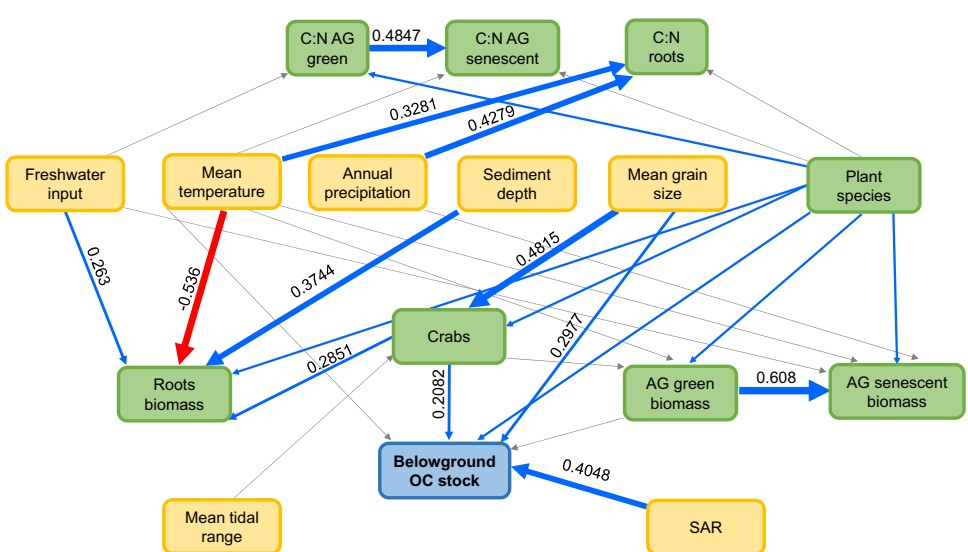

**Fig. 2 | Pathway analysis identifying biotic and abiotic drivers of organic C stocks.** SEM (structural equation model) analysis results carried out to explain belowground organic carbon stocks (sediment and roots) variability found in southern southwest Atlantic salt marshes. Green boxes correspond to biotic variables and yellow to abiotic. Blue arrows indicate significant positive effect and red arrows negative. Gray arrows are nonsignificant effects retained by the model. Numbers next to the arrows (estimates) are standardized path coefficients. The width of the arrows reflects the strength of the according pathway. OC organic carbon, AG aboveground, SAR sediment accretion rate, C:N ratio between carbon and nitrogen (Source data are provided as a Source Data file).

The values found in our study, ranging from 5.74 (SE = 0.2) to 108.81 (SE = 0.71) Mg OC ha$^{-1}$, are among the lowest reported. When we recalculate the global average pooling our results with the data from the publications mentioned above, we update the estimate to 185.88 Mg OC ha$^{-1}$. These changes in the global average, as new information appears, highlight the high variability in marsh blue C around the world and caveats the use and extrapolation of global averages. In the case of the salt marshes from southern SW Atlantic, using a global average of 317 Mg OC ha$^{-1}$ could lead to a large overestimation of the stocks. In fact, a recent publication using a literature compilation that includes different sources of data and indirect methodologies to calculate the blue C in SW Atlantic salt marshes provides an average of 270 Mg OC ha$^{-1}$ [10], which is far higher than the average we report here using actual data.

Sediment accretion rates for the marshes studied here range between 0.53 mm yr$^{-1}$ and 2.86 mm yr$^{-1}$, with a mean of 1.16 mm yr$^{-1}$. These rates are substantially lower than those reported in a global review of salt marshes, which indicates a range of 2–10 mm yr$^{-1}$, and a median value of 5 mm yr$^{-1}$ [42], suggesting lower sedimentary inputs to southern SW Atlantic marshes than marshes in other regions.

OC sequestration rates in the marshes were also comparatively low, varying between 7.38 gOC·m$^{-2}$·yr$^{-1}$ in the lower marsh of Playas Doradas and 204.21 gOC·m$^{-2}$·yr$^{-1}$ in the lower marsh of Río Negro with a mean of 47.62 gOC·m$^{-2}$·yr$^{-1}$. Estimates of global mean OC sequestration rates in marshes vary between 212 (SE = 18) gOC m$^{-2}$ year$^{-1}$ as reported by Alongi[42] (minimum of 9 and maximum of 1713 g C m$^{-2}$ year$^{-1}$) and 244.7 gOC m$^{-2}$ yr$^{-1}$ as reported by Ouyang & Lee[11] (summarizing data from 143 sites). The summary of OC sequestration rates provided by Alongi[42] is derived from 168 sites in Europe, North America, Asia, and Australia, but notably with only 8 sites in Australia and no South American or African sites included. Alongi[42] suggested that insufficient data existed from East Asia, the Arctic and Australia to obtain sound global estimates, and we further note that data is largely missing from the southern hemisphere.

### Filling geographical gaps and adding understanding behind blue C in salt marshes

Within the climate change context, there is a need to undertake carbon accounting at a national level and identify the main sources and sinks of carbon. The identification of ecosystems and areas that provide high rates of OC sequestration is essential for the development of effective measures and policies for climate change mitigation. Likewise, understanding the factors that regulate OC stocks and sequestration rates is capital for the adoption of strategies that can maximize blue C reservoirs and, thus, adaptation to climate change. This study provides a comprehensive characterization of OC stocks and sequestration rates for southern SW Atlantic salt marshes. It also highlights the importance of incorporating both environmental (i.e., abiotic) and biological (i.e., biotic) variables for an understanding of blue C storage.

Findings from this study also show the substantial global variation in blue C stocks and burial rates, as well as the large data gaps in the southern hemisphere, particularly in South America and Africa. With the inclusion of our data, it seems that overestimations in global blue C stocks and burial rates are likely to have been made. Indirect calculations derived from equations that do not incorporate local factors can be a large source of bias. For instance, in the case of the SW Atlantic salt marsh to account for the space occupied by crab burrows is key to avoid overestimates. Thus, greater effort should be made toward the assessment of blue C stocks and burial rates in understudied regions. Furthermore, reporting on aboveground biomass, which appears to not be accounted for in many studies, should be included. Our results show that in some areas, this can be a significant proportion of the total C stocks (10%).

## Methods

### Study sites

SW Atlantic salt marshes are dominated by three plant species: *Spartina alterniflora*, *Spartina densiflora* and *Salicornia* sp. Depending on the area, one, two, or all the dominant species can be found. The southern limit of *Spartina* species is 43ºS, below that, *Salicornia* sp. dominates. Where both *Spartina* species coexist, *S. densiflora* occupies the upper intertidal area while *S. alterniflora* dominates in the lower area, where it is subjected to daily tidal inundation. The mean tidal range among the studied sites varied from 0.78 to 8.07 m (micro- to macro-tidal). The dominance and presence of one or another species is mostly explained by the freshwater input[48]. Areas receiving a higher freshwater contribution are dominated by *S. alterniflora*, while those that are more saline are dominated by *Salicornia* sp.

A conspicuous characteristic of these salt marshes and the adjacent bare intertidal habitats is that they are densely inhabited by the burrowing crab *Neohelice* (=*Chasmagnathus*) *granulata*. Crabs' burrow density can reach over 120 burrows per m$^2$ and they can be up to 1 m depth[42]. This crab species can remove up to 2.4 kg of sediment per day per m$^2$ [49]. The impacts of this on ecological functions in SW Atlantic salt marshes is either by herbivory, consuming large amounts of plant biomass, or via bioturbation through the construction and maintenance of their burrows[34].

Most of the salt marshes in South America are located on the Atlantic coast, with over 95% of the recorded area of the SW Atlantic salt marshes located in Argentina (209,056 ha from 218,964)[14,22]. In this study, we sampled 11 salt marshes covering 132,796 ha spanning along ~3000 km of coastline (Fig. 1, Table 1).

### Samples collection and processing

Samples were collected in January 2016, during the southern hemisphere summer period and processed following the Blue C Manual recommendations[50], as detailed below. Sediment and vegetation samples were collected at each of the eleven salt marshes. When more than one of the dominant plant species was present in a site, sampling was undertaken in each plant community. The sampling consisted of 3 to 5 units (25 × 25 cm quadrats) per site and dominant plant species. Above and belowground biomass and OC (organic carbon) stocks, crab burrow diameter and abundance, surficial (2 cm depth) sediment for C and N stable isotope signatures and OC stocks belowground up to 1 m depth were measured at each sampling unit.

To measure aboveground plant biomass, all the biomass above the sediment within the sampling unit was collected and transported to the lab, where the material was separated into green and senescent. Samples were dried at 60 °C and weighed. A subsample of each sample was separated to determine C and N content and for stable isotope analysis. Samples of root biomass were taken using a Russian corer down to 50 cm deep and 9.05 cm$^2$ area and transported to the lab, where they were sieved. All roots were separated, dried in an oven at 60 °C and weighted. Subsamples were also processed for C and N content and stable isotopes.

To determine belowground OC stocks, up to 1 m depth samples were collected using a Russian corer. Sample depths varied between sites according to sediment hardness, reaching 1 m or hard rock were the limits. Considering this natural soil limit, we do not extrapolate the stock up to 1 m depth so as not to overestimate; rather we considered the limit for C accumulation. Each sample was sliced from top to bottom at 5, 10 and then every 10 cm up to a maximum of 1 m. All the slices were dried in at 60 °C and weighed. Subsamples of each dried slice (~2gr) were weighed, incinerated at 450 °C for 8 h and reweighed. All sediment samples (incinerated and non-incinerated) were sent to the Dauphin Island Sea Laboratory (Alabama, USA) for determination of total C and N. Incinerated samples are considered to have only inorganic C while non-incinerated have both organic and inorganic (loss on ignition (LOI) technique). Therefore, organic C content was

determined as the difference between the non-incinerated and incinerated.

To determine the contribution of allochthonous and autochthonous organic matter sources to the sediment OC stocks, C and N stable isotopes from potential sources and surficial sediment were used. To determine stable N and C isotopes in surficial sediment, we took 2 cm depth samples using a 2 cm diameter plastic tube. Samples were dried at 60 °C and then divided into two subsamples. One half was acidified to eliminate carbonates and to measure stable isotopes of C, and the other half was not acidified and used to determine stable isotopes of N. To determine C and N stable isotopes from autochthonous sources, vegetation (green and senescent leaf and roots) subsamples were ground to a fine powder and stored in Eppendorf vials. Sediment and vegetation samples were sent to the Stable Isotope Geosciences Facility, Texas A&M University (Texas, USA) for C and N stable isotope determination. Stable isotope signatures of allochthonous sources (i.e., particulate organic matter (POM) and macroalgae) were obtained from the literature (Supplementary Information Table. 3). When information from a specific site was not found, we used averages of regional data. Most of the sites have two main potential sources (salt marsh plants and POM), with the exception of Bahía San Antonio and Puerto San Julián, where macroalgal beds were also observed and reported in the literature.

To determine crab burrow abundances, all burrows present in the sampling units were counted after harvesting aboveground biomass and before sampling sediment and belowground biomass. In addition, the diameter of five burrows at each sampling unit was measured with a caliper.

A separate series of two cores were also collected from each field site to undertake a geochronological assessment of the sediment to determine accretion and OC burial rates. In the field, a walkover survey was conducted to identify areas with representative stratigraphy in the lower and upper marsh areas. Sections of the sites with minimal bioturbation were surveyed by inserting a 75 mm diameter 50 cm long core into the sediment, making sure to limit subsurface compaction (<10%[51]). Cores were sealed and packed top and bottom for transport to the lab to limit disturbance of the sediment stratigraphy[52]. In the lab, cores were removed and assessed for compaction (no compaction or disturbance recorded), and then core extremities were cleaned laterally to prevent downcore mixing and contamination and then cut into 1 cm subsamples, weighed, dried at 40 °C to constant weight and the reweighed, to assess water loss (used in the dating equations, 7). Subsamples were placed in calibrated dimension vials and analyzed in an ultra-low background HP-Ge gamma spectrometer with a mean count time of 500,000 s. Radionuclides used were $^{210}Pb$, $^{214}Pb$ (as a proxy for background $^{210}Pb$, as per ref. 50) and $^{137}Cs$ for independent verification[53]. Gamma spectroscopy is a well-established technique to assess rates of sediment accretion and carbon sequestration in coastal wetland soils[5,7,51,53–55]. To evaluate sediment accretion rates, the robust $^{210}Pb$ Constant Flux:Constant Sedimentation (CF:CS) method was used and compared with the $^{137}Cs$ impulse dating method, based on deposition from the global peak around 1963 in the southern hemisphere[52,56]. CF:CS was used to determine accretion rates for all sites but Punta Loyola given $^{210}Pb$ in excess was not separated from $^{214}Pb$ (Supplementary Fig. 1). In this case $^{137}Cs$ was used, showing a peak at 45 mm depth. In addition, the samples from Mar Chiquita were so bioturbated that dating was not possible. Carbon sequestration rates were calculated using the rates of sediment accretion and density as per Greiner et al.[5].

Following loss on ignition, minerogenic samples were analyzed for granulometry. Prior to analysis, samples were carefully disaggregated using a pestle and mortar, then mixed with sodium hexametaphosphate (to prevent flocculation), and placed on a shaker for 30 min to further disaggregate the sediments. Three runs were undertaken using a Malvern Mastersizer 2000 laser particle size analyzer with an ultrasonic prior to each run to obtain an average grain size value for each sample[52]. Samples were classified using the Wentworth scale.

## OC stocks calculation

To determine OC stocks in the aboveground and root biomass, the OC content determined in each vegetation compartment (green, senescent and roots) was multiplied by their respective biomasses measured at each site. Total belowground OC stocks were estimated by adding the OC determined along each sediment profile and expressed as mega grams (tons) per hectare (MgOC·ha). These measurement units are widely used in blue C studies, facilitating comparison. Empty space occupied by crab burrows was considered in the calculation to not overestimate OC stocks. To undertake this, the average number of burrows and diameters measured per site were used to calculate the sediment volume without C.

## Statistical analysis

The relative contribution of allochthonous (POM and macroalgae) and autochthonous (salt marsh plants) sources to the OC stocks was assessed using Stable Isotope Bayesian mixing models (MixSIAR: Stock et al. 2018, R package version 3.6). The models were run using the $\delta^{13}C$ and $\delta^{15}N$ signatures of the sediment and the 3 potential organic matter sources. The contribution of salt marsh plants was analyzed using the pooled isotopic data from green and senescent leaves and roots. The mean and standard deviations of isotopic signatures for the three sources were obtained from the literature, and concentration dependence was not incorporated into the models. Results of the mixing models are given as a probabilistic contribution (%) of each source to the sedimentary organic matter pool (mixtures).

Given that belowground OC stocks likely depend on many interrelated biotic and abiotic variables[14], we evaluated their relative importance as well as their direct and indirect pathways. Briefly, we used path analysis (a form of structural equation modeling without latent variables) to evaluate the direct and indirect roles of key abiotic and biotic factors on belowground OC stocks. Abiotic variables include grain size, tidal amplitude, mean annual temperature, freshwater inputs, mean annual precipitation, sediment accretion rates and sediment layer depth. Biotic variables include plant litter, above and belowground biomass and quality (C:N ratio), crab densities, and dominant plant species. We described the theoretical background behind each potential link in Supplementary Table 4.

Most abiotic variables were determined from external data sources at the site level. We obtained temperature and precipitation data using WorldClim (version 2; http://www.worldclim.org/)[57] through bioclimatic variables 1 (mean annual temperature, °C) and 12 (mean annual precipitation, mm). Mean tidal amplitude (m) was obtained from the Argentinean National Hydrographic Service (http://www.hidro.gov.ar/), using the information from the nearest site to our sampling locations. Freshwater input was coded as a binary variable, which indicates whether the site is associated with freshwater input from rivers (1) or not (0), partly determined following Isacch et al.[48]. All other variables were obtained in situ and at the plot level (except grain size and sediment accretion rates for which we took one measurement per dominant species with a maximum of two species per site). Grain size was estimated as the inverse of the sorting coefficient (thus showing larger values for poorly sorted sediments, typical of sediments with larger mean grain size). Plant biomass and C:N ratio (senescent and green aboveground tissues and roots) were determined as explained above and expressed as g m$^{-2}$ and unitless, respectively. We also estimated the area occupied by burrows by measuring the diameter of up to five burrows per plot, calculating the average surface of those burrows, and multiplying this average surface by the number of burrows per square meter (i.e., m$^2$ burrows per m$^2$ sediment; equivalent to the proportional surface area occupied by

crab burrows; unitless). Sediment layer depth (cm) was estimated as the maximum depth we could bury the Russian corer to obtain sediment samples. Finally, belowground OC stock (g OC m$^{-2}$) was obtained by multiplying OC density by sediment surface without burrows (see above), including OC from sediment and roots. (Source data are provided as a Source Data file).

For our path analysis, we performed a three-step procedure: (1) we first specified the metamodel carefully checking that all potential relationships were supported by scientific evidence, (2) then we simplified the individual models of each dependent variable following Deguines et al.[58], and (3) finally we performed a step-wise deletion and addition of variables that improved overall model fit and that were originally considered in the metamodel. All individual models from the metamodel included site as a random factor and thus were performed using linear mixed-effects models (lme function from the nike package for R) to evaluate the significance of the hypothesized pathways explaining each dependent variable. To ensure normality of model residuals, we transformed sediment OC stock, belowground biomass and the three C:N ratios (senescent and green aboveground tissue and roots) using log$_{10}$, and the surface of crab burrows, litter and aboveground biomass using square root. For each model, we graphically assessed the variance of the residuals for signs of heterogeneity. When heterogeneity was suspected (senescent tissue C:N), we refit the model with different variance structures and selected the structure that yielded the best results according to Akaike's information criterion[56]. We simplified each model by removing terms lacking support ($P > 0.05$) in this preliminary analysis. We performed marginal F-tests with univariate analysis of deviance[57] to investigate the effects of explanatory variables in each model (always coincident with Akaike's information criterion)[56]. Following these preliminary results, we conducted piecewise structural equation modeling to join the multiple models into a single SEM. In this implementation, individual models are constructed separately for each predicted variable. The mixed-effects models were then combined and tested with the d-separation test for goodness of fit[58] and path coefficients extracted from models that fit, as well as potential paths that were missing from the model. We implemented the model in R using the package piecewiseSEM[59]. We only deleted paths that were not significant and whose deletion improved model fit. In addition, we only added paths that were suggested by Shipley's test and that were originally considered in our metamodel.

## Data availability
Source data are provided with this paper. Organic carbon stocks, plant biomasses and crab burrow abundances and diameters data generated in this study, as well as all the data compiled for the SEM analysis and to update global estimates of OC stock and burial rate have been deposited in the Figshare database under the code: https://doi.org/10.6084/m9.figshare.24260338.

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

## Acknowledgements

We thank L. Linn from Dauphin Island Sea Laboratory, E. B. Roark from the Stable Isotope Geoscience Facility (TAMU) for laboratory analysis services and D. Navarro from IIMyC for technical and field assistance. This study was funded by grants from CONICET, FONDECyT and UNMdP (Argentina) to P.M. and O.I., NOAA (Office of Sea Grant, U.S. Dept. of Commerce, under Mississippi Alabama Sea Grant Consortium) and Mississippi State University Extension Service to E.S. Dr. N. Marbà was supported by OBAMA-NEXT, "Observing and mapping marine ecosystems - Next generation tools" project funded by the European Union under the Horizon Europe programme (grant Agreement: 101081642). The present research was carried out within the framework of the activities of the Spanish Government through the "Maria de Maeztu Centre of Excellence" accreditation to IMEDEA (CSIC-UIB) (CEX2021-001198). Dr. Raymond Ward was supported in this study by the Rising Stars Award, within the University of Brighton and by P190251PKKK to support the project "The impact of global environmental changes on coastal ecosystem services".

## Author contributions

P.M., J.A., J.C., O.I., N.M., D.M., E.S. and R.W. conceptualized the project; P.M., J.A., J.C., N.M., D.M., E.S. and R.W. carried out the fieldwork and processed the samples; J.A. and D.M. carried out the SEM analysis; M.E.B. carried out the mixing model analysis (MixSIAR); R.W. analyzed the SAR (sediment accretion rates) and the CBR (carbon burial rates); P.M. and J.A. made the figures; P.M. lead the project and wrote the manuscript which was reviewed, edited and approved by all authors.

## Competing interests

The authors declare no competing interests.
