## [Peer Review File · Nature Communications]

The Blue Carbon of southern South West Atlantic salt marshes and their biotic and abiotic driversReviewers' comments:

Reviewer #1 (Remarks to the Author):

The manuscript focuses on evaluating Blue C stocks in Argentinian saltmarshes, addressing important gaps in global carbon accounting. However, it is a descriptive, incremental study with more local significance for managing Blue C ecosystems and climate mitigation rather than a global impact. The title and abstract are misleading, implying a broader study area than just Argentina, while also omitting mention of the Malvinas/Falklands. This creates issues in acknowledging the dimension of Blue C ecosystems in the Southern Hemisphere, as noted in the introduction. The term "SW Atlantic" should be replaced with "Argentina" throughout the manuscript to accurately represent the study area. While the work provides an essential dataset for national Blue C ecosystem management, it lacks novelty (e.g., no hypothesis testing, only descriptive work). Considering this, the authors might consider submitting to a different journal, such as Scientific Reports, that aligns better with the study's focus and significance.

Title: The title could be more specific to the actual region of the study, i.e., Argentina. Additionally, the use of biotic and abiotic drivers encompasses a broad range of subjects. Please consider a more concise title that summarizes the research conducted.

Abstract: Avoid using "SW Atlantic" if the study is focused only on Argentina, as it is not accurate. Clarify whether all listed drivers were evaluated or only those significantly impacting the data. The last sentence should be corrected to clearly indicate that it refers to Argentina.

Lines 481-484: The authors mention McCowen et al. (2017), which suggests that Argentina has 68% of South America's total saltmarshes, not 95% as stated in this paper. Another recent review with a broader scope, including mangroves and seagrasses, highlights the presence of saltmarshes not only in Uruguay and South and Southeast Brazil but also in the Amazon region. The authors should acknowledge this and consider that the mapping of saltmarshes remains poorly defined, making it difficult to distinguish them from other contiguous coastal vegetated ecosystems.

Lines 503-512: Provide more information about the sampling method and core details. Were all cores corrected for decompression, and if so, how was this performed? Clearly state the core lengths and the depth at which stocks were calculated. If cores were shallower than 1 m, explain how the stocks to the top 1 m were estimated. Consider including a supplementary table with core information, including lengths, compaction, lat/long coordinates, etc.

Lines 534-538: Clarify how the authors identified areas with representative stratigraphy and provide a definition of "representative stratigraphy."

Lines 538-539: Specify the sampling method used in this context, such as a Russian corer for stocks or PVC tubes. Also, explain how compaction was limited, considering its dependence on soil type and sampling device.

Line 564: Please specify how you estimated stocks up to 1 m (if you did) if the cores did not reach that depth. Clearly state the depth to which the data discussed in the paper are referred to. Please present this in a table in the SI.

Line 584: In the Path analysis, Line 589 mentions that the authors used freshwater inputs, ppt, temperature, etc. Clarify the sources from which this data was obtained and ensure that all relevant metadata is included in the supplementary material. Present ALL the numbers used in the statistical analysis in a table for better clarity.

Line 594: Table SI4 only provides the rationale for using the data but does not show the actual data values. Amend this table to include the values used for each region so that readers can observe the variability of these data, as explained above. Also, clarify if the resolution of the data was appropriate and available for each studied region.

Line 601: State the number of cores that were dated and present the Pb-210 and Ra profiles in the supplementary information. This is crucial due to the scarcity of data for the region. Explain whether all cores used for the determination of accumulation rates. Address how the presence of crabs affected core mixing and provide relevant details.

Line 613: Clarify if reaching "hard rock" was the issue, or if the type of core used could not penetrate hard-packed soil or soil rich in roots, for instance.

Line 81: The values I get from Mcowen et al for Argentina is 118,870 ha which is different from the value presented here, can you please check?

Line 82: Clearly specify that you are only working with saltmarshes from Argentina. Acknowledge the existence of these ecosystems in Uruguay and Brazil, even though data for the majority of the Brazilian and Uruguayan coast might be scarce or less abundant. Discuss the potential impact of better mapping of saltmarshes on the dominance of Argentina in the occurrence of saltmarshes and its consequences for global estimates.

Line 109: Provide a more thorough explanation of how the Pb-210 profiles were evaluated and how they varied along cores and between sites to better assess accretion rates. Add the profiles in the SI.

Lines 118-126: Without looking at the models, it is very hard for the reader to assess the OM source apportioning. The models (and uncertainties) need to be presented at least in the supplementary material. I wonder how was the fit. Besides, a table presenting the end-member values should also be provided.

Lines 155-159: The authors ignored to mention the data on Falklands (e.g. Payne et al., 2019). Compare the generated data with all available literature for the region, including data from the Falklands (e.g., Payne et al., 2019).

Lines 164: Note that the reference Hatje et al., 2023, truly worked with the whole SW Atlantic, i.e., 0-55°S, and included data from the Falklands. Explore the reasons for the differences in results and contextualize the data used for comparison, considering the uncertainties associated with extrapolations.

Lines 166-170: Discuss the region of the Falklands, which was not mentioned previously, and provide relevant information about it.

Lines 196: Include additional references at this point in the manuscript.

Line 217: Change "SW Atlantic" to "Argentinian."

Line 243: Change "SW Atlantic" to "Argentinian."

Reviewer #2 (Remarks to the Author):

The Blue Carbon of South West Atlantic salt marshes and their biotic and abiotic drivers

This paper addresses an important topic and does a valuable service in fulfilling knowledge gaps on blue carbon in South America. Furthermore, it also contributes to the understanding of the bioturbation processes in the carbon cycling. However, I think that there are several sections within the manuscript that could be improved before publication.

Major comments:

1. The authors provide an updated global estimate of saltmarsh stocks and burial rates. I consider this as a KEY contribution of their manuscript. However, no clear information is provided on how these estimates were calculated and these results are loosely presented in the discussion.
2. The study is framed around blue carbon in the SW Atlantic Ocean, but the study is focused on Argentina. This can be misleading/confusing for readers not familiar with South America. Even though the authors mention that the majority of saltmarshes is located in Argentina, there are some major improvements needed in the text and Figure 1 (or the addition of a new figure) that could help clarify these points.
3. Introduction and Discussion need to provide further context on blue carbon and saltmarshes within the SW Atlantic coast more broadly.

Specific comments:

1. Introduction: saltmarsh BC accounting is also incomplete due to major gaps in their distribution. A new study has been released recently (details on my comment about Figure 1) is trying to minimise

these uncertainties but would be interesting if the authors could also include a few sentences on the topic for South America. What is the state of knowledge of BC and saltmarshes within Argentina and other countries located along the SW Atlantic coast?

2. The role of bioturbation in BC cycling is also an important contribution from the paper. Since biological processes are usually excluded from studies due to their complexity, I wonder if the authors could consider adding an infographic showing the expected interactions affecting C transformation pathways in salt marshes of the SW Atlantic coast?

3. The authors need to clarify within the text soil carbon is being combined with belowground carbon. This info is only available in the legend for Table 1, and gets confusing since Figure 1 shows the data separated by roots and soil. My preference is to keep belowground and soil pools separately to avoid confusion and transparency in the accounting for different carbon pools. In this case, the modifications need to be made throughout the manuscript.

4. The representation of SE throughout the text needs to be standardized. There are a few occasions where just a number is provided within () and others where SE= is used. I recommend the use of the latest throughout to make it easier for the reader.

5. The recalculation of carbon stocks and sequestration rates based on the new information provided by this study is loosely provided in the discussion. I argue that this is a key contribution of this study and could be moved to the results. Regardless of the authors' decision, more information should be provided on how these calculations were done, including the baseline information used (e.g., saltmarsh area). It would be interesting to add a table comparing the previous and new estimates for stocks and sequestration rates.

6. The authors mention an Extended Data Table 1a but this table is not available in the submission package. I'm assuming this table is the one provided in the excel file, which is nominated Extended Data Table 3a.

7. Discussion: the authors mentioned 'OC stocks in soils have been estimated to hold an average of 99.2% of the total stock, although the data in this study show a lower percentage (84.9%).' However, in the results the authors claim 'Altogether, the 11 salt marsh sites in this study contain 4,136,190 tons of OC of which 3,928,493 tons were found belowground (95%, Table 2, Fig. 1).' I'm assuming this difference in the % is due to the combination of soil and roots as 'belowground' C but would be good to have further clarification.

8. I understand that there are some limitations on the blue carbon studies conducted in South America but would be interesting if the authors could explore/discuss this more broadly in the discussion.

9. The authors claim 'This study provides a comprehensive characterization of OC stocks and sequestration rates for SW Atlantic salt marshes.' However, the study is focusing only on Argentina's saltmarshes. I provided a few comments to help clarify this throughout the manuscript, but maybe the authors could also consider adding a sentence here in this 2nd last paragraph.

10. Figure 1: Add a new panel or a new figure showing the entire SW Atlantic region and the saltmarsh distribution within the region. If local/regional maps for Argentina/Uruguay/Brazil are not available, the authors could consider the new saltmarsh map released this year.

Below you will find our responses (*in italic*) point by point to all the reviewers's comments:

Reviewer #1 (Remarks to the Author):

The manuscript focuses on evaluating Blue C stocks in Argentinian saltmarshes, addressing important gaps in global carbon accounting. However, it is a descriptive, incremental study with more local significance for managing Blue C ecosystems and climate mitigation rather than a global impact. The title and abstract are misleading, implying a broader study area than just Argentina, while also omitting mention of the Malvinas/Falklands. This creates issues in acknowledging the dimension of Blue C ecosystems in the Southern Hemisphere, as noted in the introduction. The term "SW Atlantic" should be replaced with "Argentina" throughout the manuscript to accurately represent the study area. While the work provides an essential dataset for national Blue C ecosystem management, it lacks novelty (e.g., no hypothesis testing, only descriptive work). Considering this, the authors might consider submitting to a different journal, such as Scientific Reports, that aligns better with the study's focus and significance.

We strongly disagree with the reviewer in this major point; we believe that this is not just "a descriptive incremental study". First of all, it fulfills a major gap in information previously acknowledged by several studies (see references 11 and 25 as examples). Besides the semantic used to name the study area, our study embraces most salt marshes occurring in the SW Atlantic coast, and certainly the largest ones. Moreover, we are dealing with ecosystems that don't recognize national borders, and it happens to be the case that most of them occur within the Argentinean borders (see lines 79-85). Second, the Malvinas's information recommended by the reviewer doesn't belong to a salt marsh (it's a peatland) so we still consider that it shouldn't be mentioned in this study about salt marshes (we recall this issue below). Third, and regarding the global impact of our results, it is important to mention that our data leads to major changes in global estimations of C stocks and burial rates. Fourth, we go beyond this by explicitly testing the impact of different biotic and abiotic drivers. In particular, the hypothesis that biotic interactions could be driving the Blue C in the region has been published by part of this group before (Martinetto et al. 2016, reference 15) and here was specifically tested at a large scale. So far, biotic drivers were generally set aside when estimating drivers of C stocks at large scales. All the rationale that justifies the incorporation of the selected variables is provided to support the hypotheses tested. Moreover, in the introduction (lines 57-71) we provide the context that leads to carry out this study. Based on all the above we believe that this manuscript is indeed a novel contribution with consequences for global estimates and to understand the processes governing carbon stocks. We understand that part of the reviewer's confusion could be due to the writing so we reinforced in this new version all these issues as explained in the points below.

Title: The title could be more specific to the actual region of the study, i.e., Argentina. Additionally, the use of biotic and abiotic drivers encompasses a broad range of subjects. Please consider a more concise title that summarizes the research conducted.

*The title was changed to: "The blue C of **southern** SW Atlantic salt marshes and its biotic and abiotic drivers". This title specified that the study refers to the southern part of the SW Atlantic coast; however, we still think that covering the 70% of the total area reported for salt marshes in the Atlantic coast in one study is enough reason to avoid the use of national border to limit natural systems. Regarding the drivers, the limitation in title words impede to mention driver by driver, so after discussing this we several colleagues we decided to keep biotic and abiotic.*

Abstract: Avoid using "SW Atlantic" if the study is focused only on Argentina, as it is not accurate. Clarify whether all listed drivers were evaluated or only those significantly impacting the data. The last sentence should be corrected to clearly indicate that it refers to Argentina.

All the text has been edited according to the new title.

Lines 481-484: The authors mention McCowen et al. (2017), which suggests that Argentina has 68% of South America's total saltmarshes, not 95% as stated in this paper. Another recent review with a broader scope, including mangroves and seagrasses, highlights the presence of saltmarshes not only in Uruguay and South and Southeast Brazil but also in the Amazon region. The authors should acknowledge this and consider that the mapping of saltmarshes remains poorly defined, making it difficult to distinguish them from other contiguous coastal vegetated ecosystems.

*The Reviewer is right, according to McCowen et al. 2017 Argentina has 68% of the South American salt marshes. However, that paper didn't include the southern Patagonian saltmarshes that were well described by Isacch et al. 2006 and Bortolus et al. 2009 and compiled by Martinetto et al. 2016. In addition, in this version we emphasize that we specifically refer to southern SW Atlantic salt marshes, and 95% of them are located in Argentina according to the sources cited (lines 79-85). The text now reads: "Most of the salt marshes in South America are located on the Atlantic coast with over 95% of the surface of these Atlantic marshes located in Argentina (209,056 ha from 218,964)^{15,23}. In this study we sampled 11 salt marshes covering 197,577 ha of which 145,309 ha are dominated by strictly salt marsh species (*Spartina densiflora*, *Spartina alterniflora* and *Sarcocornia* sp.) and 52,268 ha by brackish water species¹⁵. All the results showed here correspond to the area dominated by salt marsh species. These sites span along ~3000 km of coastline (35° to 51° S, Fig 1)."*

Lines 503-512: Provide more information about the sampling method and core details. Were all cores corrected for decompression, and if so, how was this performed? Clearly state the core lengths and the depth at which stocks were calculated. If cores

were shallower than 1 m, explain how the stocks to the top 1 m were estimated. Consider including a supplementary table with core information, including lengths, compaction, lat/long coordinates, etc.

Sampling was done using a Russian corer which is recommended by the Blue C Manual specifically because it minimizes soil compaction Core depth is already in Table 1 and all the procedure is described in lines 489-581. All the information required by the reviewer can be found in tables 1 and 2 and in extended and supplementary data.

Lines 534-538: Clarify how the authors identified areas with representative stratigraphy and provide a definition of "representative stratigraphy."

Done. It was clarified in lines 542-545

Lines 538-539: Specify the sampling method used in this context, such as a Russian corer for stocks or PVC tubes. Also, explain how compaction was limited, considering its dependence on soil type and sampling device.

Done. See answer above.

Line 564: Please specify how you estimated stocks up to 1 m (if you did) if the cores did not reach that depth. Clearly state the depth to which the data discussed in the paper are referred to. Please present this in a table in the SI.

We clarified the procedure in lines 507-511, and the information required can be found in tables 1 and 2 and in extended and supplementary data.

Line 584: In the Path analysis, Line 589 mentions that the authors used freshwater inputs, ppt, temperature, etc. Clarify the sources from which this data was obtained and ensure that all relevant metadata is included in the supplementary material. Present ALL the numbers used in the statistical analysis in a table for better clarity.

Sources of information for all variables that were specifically sampled in this study were specified, see lines 594-616.

We know that data should be public for this journal so we included a new extended data table (Extended Data SEM_data) with ALL the numbers used in the SEM analysis that can be published along with the paper if accepted.

A recent paper published in Nat Comm by part of our group (Nat Commun 14, 1809 (2023). <https://doi.org/10.1038/s41467-023-37395-y>) include a similar SEM analysis. The way we provided the information here is based in their experience and in concordance with the journal requirement.

Line 594: Table SI4 only provides the rationale for using the data but does not show the actual data values. Amend this table to include the values used for each region so that readers can observe the variability of these data, as explained above. Also, clarify if the resolution of the data was appropriate and available for each studied region.

See answer to the point above.

Line 601: State the number of cores that were dated and present the Pb-210 and Ra profiles in the supplementary information. This is crucial due to the scarcity of data for the region. Explain whether all cores used for the determination of accumulation rates. Address how the presence of crabs affected core mixing and provide relevant details.

All this information is available in lines 534-555 and we have added the profile figures as Supplementary Information Fig. 1

Line 613: Clarify if reaching "hard rock" was the issue, or if the type of core used could not penetrate hard-packed soil or soil rich in roots, for instance.

Done. See lines 507-508

Line 81: The values I get from McCowen et al for Argentina is 118,870 ha which is different from the value presented here, can you please check?

We used two sources of data given that McCowen did not include the southern Patagonian salt marshes which are vast and we have included them in our study, that is the reason of the difference in hectares reported. The two references are given in the text, so we never said that we get the information exclusively from McCowen et al. 2017. Moreover, the text specified that most of the South American salt marshes are in the Atlantic coast, and that 95% of these salt marshes in the Atlantic coast of South America are in Argentina. We have edited this paragraph (lines 79-85) anyways to avoid confusions and modified Fig 1 for clarification.

Line 82: Clearly specify that you are only working with saltmarshes from Argentina. Acknowledge the existence of these ecosystems in Uruguay and Brazil, even though data for the majority of the Brazilian and Uruguayan coast might be scarce or less abundant. Discuss the potential impact of better mapping of saltmarshes on the dominance of Argentina in the occurrence of saltmarshes and its consequences for global estimates.

See the answer above.

Line 109: Provide a more thorough explanation of how the Pb-210 profiles were evaluated and how they varied along cores and between sites to better assess accretion rates. Add the profiles in the SI.

Done. See lines 534-555. The profiles were added as SI.

Lines 118-126: Without looking at the models, it is very hard for the reader to assess the OM source apportioning. The models (and uncertainties) need to be presented at least in the supplementary material. I wonder how was the fit. Besides, a table

presenting the end-member values should also be provided.

This information can be found in lines 574-783 and Supplementary Information Tables 1 and 2.

Lines 155-159: The authors ignored to mention the data on Falklands (e.g. Payne et al., 2019). Compare the generated data with all available literature for the region, including data from the Falklands (e.g., Payne et al., 2019).

The study by Payne et al. 2019 (Peatland initiation and carbon accumulation in the Falkland Islands. Quaternary Science Review 212:0213-218) was done in a peatland so their results are not comparable with ours given that those are ecosystems completely different from salt marshes. Regarding the comparison of our results with all available literature for the region please see the answer below.

Lines 164: Note that the reference Hatje et al., 2023, truly worked with the whole SW Atlantic, i.e., 0-55°S, and included data from the Falklands. Explore the reasons for the differences in results and contextualize the data used for comparison, considering the uncertainties associated with extrapolations.

As explained above, we disagree with the incorporation of the Malvinas's study because it is clearly a peatland and not a salt marsh. Peatland are recognized for preserving enormous amounts of C in their soils. Thus, the inclusion of that study in global stocks calculations in salt marshes would likely lead to an overestimation of the actual capacity of salt marshes to store C.

Moreover, even when it is true that Hatje et al. 2023 worked with the whole SW Atlantic, their study calculated C stocks from the literature, consisting in 6 articles and 1 book chapter for Argentina. However, among them, only the book chapter actually report soil OC data for a salt marsh in Península Valdés, all other sources of data report %OM determined in surficial sediment samples (some of the co-authors of this ms co-authored those papers and we all have worked on the ecology of those sites). Based in our determinations of actual C (along the entire soil profile), and also on the exploration of the relationship between OM and C found in our samples, we know that there is not a single equation to convert %OM to C for the salt marshes of the region. For instance, the conversion equations provided by the Blue C Manual do not fit to any of the salt marshes studied here. Moreover, the numbers reported in those studies were not corrected by the volume belonging to crab burrows which also potentially leads to overestimation. That is why we analyzed a large number of samples instead of using published conversion equations. Each site follows a different trajectory, so the conclusions and extrapolations made by Hatje et al. 2023 should be taken with caution. That being said, we have incorporated some lines specifically discussing this issue (see lines 46-48 and 255-258)

Lines 166-170: Discuss the region of the Falklands, which was not mentioned previously, and provide relevant information about it.

As expressed above, the Malvina's study mentioned by the reviewer does not deal with salt marshes.

Lines 196: Include additional references at this point in the manuscript.

Done. See lines 195-214.

Line 217: Change "SW Atlantic" to "Argentinian."

Line 243: Change "SW Atlantic" to "Argentinian."

This was corrected all through the ms according to the new title.

Reviewer #2 (Remarks to the Author):

The Blue Carbon of South West Atlantic salt marshes and their biotic and abiotic drivers

This paper addresses an important topic and does a valuable service in fulfilling knowledge gaps on blue carbon in South America. Furthermore, it also contributes to the understanding of the bioturbation processes in the carbon cycling. However, I think that there are several sections within the manuscript that could be improved before publication.

Major comments:

1. The authors provide an updated global estimate of saltmarsh stocks and burial rates. I consider this as a KEY contribution of their manuscript. However, no clear information is provided on how these estimates were calculated and these results are loosely presented in the discussion.

Data, methods and reference sources for global estimations are presented in Extended Data Table 1a and 1b. We have added a section highlighting this results in lines 144-155.

2. The study is framed around blue carbon in the SW Atlantic Ocean, but the study is focused on Argentina. This can be misleading/confusing for readers not familiar with South America. Even though the authors mention that the majority of saltmarshes is located in Argentina, there are some major improvements needed in the text and Figure 1 (or the addition of a new figure) that could help clarify these points.

We have clarified the geographic area of our study in the text (lines 78-82 and 469-477) and we have added a South American map as a new panel in Fig 1.

3. Introduction and Discussion need to provide further context on blue carbon and saltmarshes within the SW Atlantic coast more broadly.

Done. We have added information in lines 46-48 and 252-260.

Specific comments:

1. Introduction: saltmarsh BC accounting is also incomplete due to major gaps in their

distribution. A new study has been released recently (details on my comment about Figure 1) is trying to minimise these uncertainties but would be interesting if the authors could also include a few sentences on the topic for South America. What is the state of knowledge of BC and saltmarshes within Argentina and other countries located along the SW Atlantic coast?

Done

2. The role of bioturbation in BC cycling is also an important contribution from the paper. Since biological processes are usually excluded from studies due to their complexity, I wonder if the authors could consider adding an infographic showing the expected interactions affecting C transformation pathways in salt marshes of the SW Atlantic coast?

Thanks for this comment. We didn't add such infographic because we published that before (see Martinetto et al. 2016 reference 15). That paper specifically analyzes how crabs (through bioturbation and herbivory) could mediate the C transformation pathway in salt marshes of the SW Atlantic coast. However, in the introduction (lines 66-76) and in the discussion (lines 198-217) we expanded the text to increase their clarity.

3. The authors need to clarify within the text soil carbon is being combined with belowground carbon. This info is only available in the legend for Table 1, and gets confusing since Figure 1 shows the data separated by roots and soil. My preference is to keep belowground and soil pools separately to avoid confusion and transparency in the accounting for different carbon pools. In this case, the modifications need to be made throughout the manuscript.

For transparency, we presented the information separated. However, given that few studies separate between these two components of belowground C, we think that keeping them altogether would be more useful for global comparisons.

4. The representation of SE throughout the text needs to be standardized. There are a few occasions where just a number is provided within () and others where SE= is used. I recommend the use of the latest throughout to make it easier for the reader.

Done

5. The recalculation of carbon stocks and sequestration rates based on the new information provided by this study is loosely provided in the discussion. I argue that this is a key contribution of this study and could be moved to the results. Regardless of the authors' decision, more information should be provided on how these calculations were done, including the baseline information used (e.g., saltmarsh area). It would be interesting to add a table comparing the previous and new estimates for stocks and sequestration rates.

Done. We moved the global estimations to the results (lines 144-155) and further details on the calculations in Extended data Tables 1a and 1b.

6. The authors mention an Extended Data Table 1a but this table is not available in the

submission package. I'm assuming this table is the one provided in the excel file, which is nominated Extended Data Table 3a.

Thanks to the reviewer for identifying our mistake. *It is now corrected.*

7. Discussion: the authors mentioned 'OC stocks in soils have been estimated to hold an average of 99.2% of the total stock, although the data in this study show a lower percentage (84.9%).' However, in the results the authors claim 'Altogether, the 11 salt marsh sites in this study contain 4,136,190 tons of OC of which 3,928,493 tons were found belowground (95%, Table 2, Fig. 1).' I'm assuming this difference in the % is due to the combination of soil and roots as 'belowground' C but would be good to have further clarification.

It was clarified.

8. I understand that there are some limitations on the blue carbon studies conducted in South America but would be interesting if the authors could explore/discuss this more broadly in the discussion.

Done. We have included this point in the subsection "Filling geographical gaps and adding understanding behind blue C in salt marshes" line 239

9. The authors claim 'This study provides a comprehensive characterization of OC stocks and sequestration rates for SW Atlantic salt marshes.' However, the study is focusing only on Argentina's saltmarshes. I provided a few comments to help clarify this throughout the manuscript, but maybe the authors could also consider adding a sentence here in this 2nd last paragraph.

Done. This issue was clarified from title to discussion, including Fig .1

10. Figure 1: Add a new panel or a new figure showing the entire SW Atlantic region and the saltmarsh distribution within the region. If local/regional maps for Argentina/Uruguay/Brazil are not available, the authors could consider the new saltmarsh map released this year.

Done

REVIEWERS' COMMENTS

Reviewer #2 (Remarks to the Author):

Overall, this paper is much improved and I'm glad the authors were able to make good use of the comments provided by the reviewers. This publication will make a great contribution to the efforts to improve global C accounting while also filling several knowledge gaps within South America.

Reviewer #3 (Remarks to the Author):

OVERVIEW

The SW Atlantic salt marshes have an interesting top down control by abundant crabs. The effect of this on carbon turnover is not well described in the literature and thus this article makes an important contribution globally. The article provides a valuable dataset for southern hemisphere coastal ecosystems focusing on a holistic study for Argentina.

I was hoping to see some biogeochemistry and other detail of crab effects in terms of burrow structure and organic matter degradation. However, the authors only measured crab abundance in terms of burrows and burrow width. Nonetheless a manuscript worthy of publication.

- What are the noteworthy results?

Blue carbon data are presented over a latitudinal gradient from 74° 35' S to 51° S for southern SW Atlantic salt marshes. Most (95%) of these salt marshes occur in Argentina (209,056 ha of 218,964).

The significance of this article is not only the contribution of knowledge to this global geographic gap but rather the inclusion of bioturbation and crab burrows / burrow volume in the findings. The calculation of sediment carbon included burrow volume which possibly accounted for the much lower sediment carbon values than that reported globally. This is an important consideration for global studies where bioturbation is an issue. At the study sites the burrowing crab *Chasmagnathus granulata* causes significant disturbance with burrow density over 120 burrows per m² occurring up to 1 m depth.

- Will the work be of significance to the field and related fields? How does it compare to the established literature? If the work is not original, please provide relevant references.

The research highlights the importance of incorporating abiotic and biotic variables for an understanding of blue carbon storage. The values reported for the salt marsh are some of the lowest recorded globally and this has a significant influence on revised global estimates of blue carbon storage.

This research contributes to knowledge on bioturbation processes in carbon storage and provides opportunities for follow up research at other global sites.

Point by point response (*in italic*) to Reviewer#3:

Review: The blue carbon of southern SW Atlantic salt marshes

OVERVIEW

The SW Atlantic salt marshes have an interesting top down control by abundant crabs. The effect of this on carbon turnover is not well described in the literature and thus this article makes an important contribution globally. The article provides a valuable dataset for southern hemisphere coastal ecosystems focussing on a holistic study for Argentina.

I was hoping to see some biogeochemistry and other detail of crab effects in terms of burrow structure and organic matter degradation. However, the authors only measured crab abundance in terms of burrows and burrow width. Nonetheless a manuscript worthy of publication.

We thank the reviewer for considering our study of interest and worthy of publication. Regarding the specific effects of crabs and their burrows, this is something that we have explored at local scale in several papers through experimental field studies. Based on these studies, our experience and the logistic difficulties to manipulate crabs/burrows at the large scale included in this manuscript, we decided to use the number and size of burrow as a magnitude of the crab effects. These measurements were also used to correct the C stocks by empty space occupied by burrows.

Blue carbon data are presented over a latitudinal gradient from 35° to 51° S for southern SW Atlantic salt marshes. Most (95%) of these salt marshes occur in Argentina (209.056 ha of 218.964). The significance of this article is not only the contribution of knowledge to this global geographic gap but rather the inclusion of bioturbation and crab burrows / burrow volume in the findings. The calculation of sediment carbon included burrow volume which possibly accounted for the much lower sediment carbon values than that reported globally. This is an important consideration for global studies where bioturbation is an issue. At the study sites the burrowing crab *Chasmagnathus granulata* causes significant disturbance with burrow density over 120 burrows per m² occurring up to 1 m depth.

We thank the reviewer for the comment.

The research highlights the importance of incorporating abiotic and biotic variables for an understanding of blue carbon storage. The values reported for the salt marsh are some of the lowest recorded globally and this has a significant influence on revised global estimates of blue carbon storage.

This research contributes to knowledge on bioturbation processes in carbon storage and provides opportunities for follow up research at other global sites.

We thank the reviewer for these comments.

SUGGESTED EDITS

Abstract

Coastal vegetated ecosystems are important due to their capacity to sequester organic carbon (blue C). Yet, blue C global accounting is incomplete, with major gaps for southern hemisphere data. Datasets show a large variability suggesting that the interaction between environmental and biological drivers is important at the local scale. Here we found that southern SW Atlantic salt marshes store on average **42.43 + 27.56 Mg OC ha⁻¹ with 40.74 SE= 2.7+ 27.56 Mg OC ha⁻¹** in belowground components and bury on average 47.62 g OC m⁻² yr⁻¹ (ranging from 7.38 to 204.21). Accretion rates, granulometry, plant species and burrowing crabs were identified as the main factors in determining belowground OC stocks. These data provide to an updated global estimation for

stocks in salt marshes of 185.89 Mg OC ha⁻¹ (n= 743; SE= 4.92) and a C burial rate of 199.61 g OC m⁻²yr⁻¹ (n= 193; SE= 16.04), which are lower than previous estimates. 2
Perhaps change format of this to 47.62 g OC m⁻² + so in similar to the format in previous sentence.

SE is reported following the recommendations by “Zar, J.H. (2010) Biostatistical analysis. Fifth Edition. Prentice Hall”. Given that it is not a limit of confidence it is not correct to report it using ±.

Underline text shows where edits should be made.

Line 58 states “nutrient availability” also “herbivory”. If these are important factors to consider at your sites in future blue carbon measurements then mention as a knowledge gap in your Discussion.

Thanks for the comment. Nutrient availability was included in the SEM analysis as N content. Herbivory is included in the discussion (see lines 208-211).

Line 68 correct spelling of coastal
Corrected.

Comment: great latitudinal gradient – any significance for results related to temperature or rainfall? other biogeographic patterns?
The SEM analysis did not identify any effect of temperature or precipitation on C stocks. We expected to find some effects as we explain in the rationale for all the variables included in Table 4 of Supplementary Information.

Line 75 we carried out a path analysis using SEM (structural equation modelling).
Thanks for noting this, we have included the meaning of the abbreviation (line 77).

Line 120 Particulate
Corrected.

Macroalgae at Bahía San Antonio and Puerto San Julián; were these sites eutrophic? is this an important consideration for future studies and to be noted in the Discussion?
San Antonio and San Julián they both show blooms of macroalgae in response to large anthropogenic nutrient contributions but also with large water masses flushing twice a day so the environment is well oxygenated. We considered nutrient availability as a potential driver of blue C so we include C:N ratios in the SEM analysis. No effect was detected.

Line 156. When we recalculate the global average accumulation rate adding our results....
Corrected.

Line 161. Global estimations for salt marsh have varied as more data become available, with a current average range of 162.....
Corrected.

Lines 223-232 Growth form i.e. grass (*Spartina*) or succulent (*Salicornia*) salt marsh species, maybe important to mention this, as plant species had a significant influence on belowground OC stock.
We have included this in lines 191-193.

Lines 261 “For instance, in the case of the SW Atlantic salt marsh to account for the space occupied by crab burrows is key to avoid overestimates.” This is an important result of this research; please consider inclusion in the abstract.
We have include a sentence in the abstract (lines 29-30).

Table 1. Carb burrow abundances
Corrected.

abundance should be per unit area, check units in Tables and Figures
We have checked the units in tables and figures.

Figure 2 please add sediment to the belowground OC stock label, in order to be more descriptive;
root biomass is a separate box. Please add full descriptor for SAR so that self-explanatory i.e.
sediment accretion rate
*Belowground refers to sediment and roots; root biomass does not refer to C content, as indicated,
this is the biomass. Full description of SAR was added in the figure legend (lines 663-665).*

Line 477 Please add the tidal range.
We added tidal range in lines 274-275.

Line 485 Correct spelling of Atlantic
Corrected.

Line 487 Are the crabs equally abundant in the *Spartina* and *Salicornia* zones? Check if some
comment needed on this.
Crabs inhabit at both with no defined pattern.

Line 513 reword to... extrapolate the stocks to 1 m in order not to overestimate...
We have reworded the sentence, see lines 310-311.

Line 564 In this case
Corrected.

Line 566 so bioturbated that dating was not possible.
Corrected.

Check use of salt marsh or saltmarsh throughout; be consistent
Checked.